# Solution structure of Titan-relevant aqueous ammonia by neutron diffraction
Mazin Nasralla [1], Harrison Laurent [1], Oliver L. G. Alderman [2] & Lorna Dougan [1] ✉

In 2034, NASA Dragonfly will arrive at Titan's Selk crater to study an environment where molten ice has potentially interacted with organics. Some models suggest that Titan has a sub-surface ocean enriched in ammonia, a molecule that forms a deep eutectic with water, implying that it strongly perturbs water's intermolecular structure. In anticipation of the Dragonfly mission, and to understand the effects of the addition of ammonia to liquid water, we used neutrons to probe the structure of a 20.5 wt.% ammonia-water solution at 273 K and 298 K at 1 bar. We observed the formation of ice-like motifs in ammonia's hydration shell, a result reminiscent of the 'microscopic icebergs' predicted to form around methane and non-polar solutes that were a feature of the original hypothesis for the hydrophobic effect. This result may have implications for the aqueous chemistry of Titan and ammonia-rich ocean worlds.

In 2034, the Dragonfly probe will land near the Selk impact crater on Saturn's moon Titan, a landing site selected for 'the likely presence of exposed deposits of water-rich material, potentially including materials where molten ice has interacted with organics'[1], possibly in the presence of ammonia[2]. The significance of the interaction of ammonia with liquid water and organics was recently highlighted by laboratory analysis of material returned from the surface of the asteroid, Bennu[3]. Analysis of the Bennu sample found 'abundant ammonia', and evidence for 'soluble organics formation and alteration by low-temperature reactions, possibly in ammonia-rich fluids'[4]. The organic matter included 14 proteinogenic amino acids, amines, formaldehyde, carboxylic acids, polycyclic aromatic hydrocarbons, and the five nucleobases found in DNA and RNA[4]. The presence of the 'building blocks of Life' with evidence of water-rich fluids has significance for the origins of Life on Earth. That these organic building blocks may have been solvated by aqueous ammonia underlines the importance of understanding the effect that ammonia addition has on the intermolecular structure of liquid water and prebiotic molecules. Terrestrial biochemistry has shown that the self-assembly and aggregation of molecular building blocks is linked to the hydrophobic interaction, an interaction that was historically linked to 'quasi-solid'[5] crystalline structure in the hydration layer of volatile solutes such as methane[5], and in non-polar solute hydration shells in general[6]. In the case of terrestrial structured biomolecules, such as proteins, protein folding is driven by the totality of atomic interactions in a solvation shell of hydrogen bonded water molecules, ions and solutes[7]. The properties of Earth's liquid water that make it a 'matrix for life'[8] have been related to the strength and extent of what is the densest known hydrogen bond network of any known material[9]. The effect then of ammonia addition, a molecule capable of hydrogen bond formation, on the intermolecular

structure of liquid water is relevant to the assembly and stability of prebiotic molecules in ammonia-rich ocean worlds, and the subsequent delivery of complex organics to the inner solar system. Figure 1b describes the phase diagram of ammonia-water mixtures at 1 bar that includes a deep eutectic at 33 wt.% ammonia, 176 K[10], implying that ammonia addition significantly perturbs the hydrogen bonded structure of liquid water.

The subject of this work is the intermolecular structure of aqueous ammonia in the context of the outer solar system, and particularly the impact melts of Saturn's moon Titan. In 1972, a theoretical model was proposed for the formation of the ice-giants and their icy satellites through the low-temperature condensation of ices in thermodynamic equilibrium with the cooling gas of the solar nebula[11]. The condensation of significant quantities of solid ammonia hydrates was a key prediction of this model, such that the chemical composition of the icy satellites was estimated to be ~10% ammonia by mass[12]. The seminal work[11] led to astrobiological interest in the effect that ammonia might have in depressing the freezing point of water, a phenomenon that would extend the phase space of liquid water in ammonia-rich worlds. Much interest centred on Saturn's moon Titan, whose methane-rich[13,14] atmosphere was thought to host photolytic chemistry and a red aerosol of complex hydrocarbons that would be hydrolysed to form amino acids in the presence of ammonia in Titan's regolith[15]. In 2005, Cassini's Huygen's probe measured Titan's atmosphere to be 98% nitrogen and 2% methane[16] before landing on what appeared to be a dry lakebed at 93.7 K, 1470 mbar[17]. Observations by the Cassini spacecraft[18] characterised Titan as a volatile-rich world with methane lakes at its Northern pole[19], and a sub-surface ocean of liquid water[20] that might be enriched in ammonia[21], although other solutes such as ammonium sulphate have also been proposed[20]. Initial models of Titan's sub-surface

¹School of Physics and Astronomy, University of Leeds, Leeds, UK. ²ISIS Neutron and Muon Source, Harwell Campus, Didcot, UK.
✉e-mail: L.Dougan@leeds.ac.uk

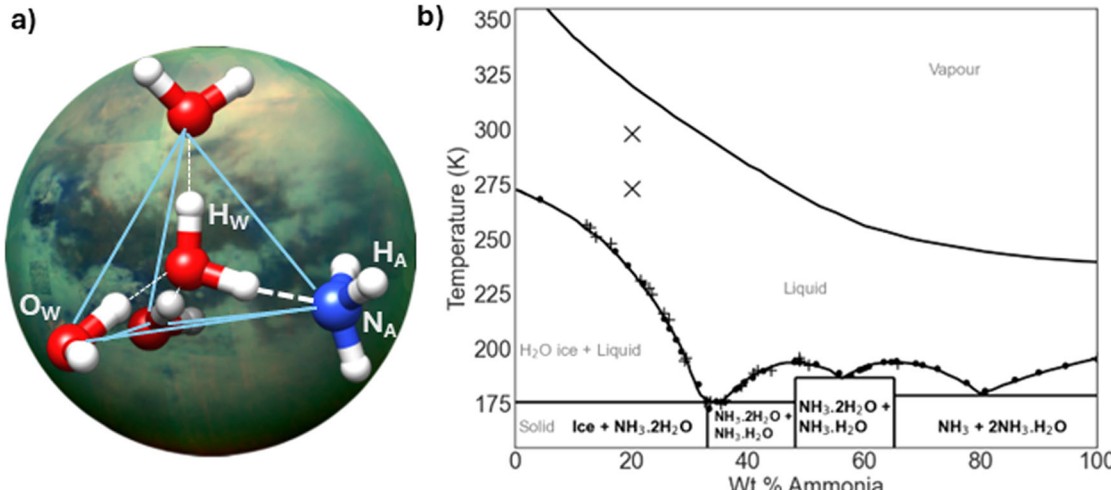

**Fig. 1 | Titan, the intermolecular structure of aqueous ammonia and its phase diagram. a** A representative ball and stick model of the tetrahedral (outlined by blue lines) solvation of a water molecule, reinforced by a water-ammonia hydrogen bond (thick white dashed line), with an image of Titan set as the backdrop (Credit: NASA/JPL). The atom labels are used throughout the results that follow. **b** The phase diagram of ammonia-water at 1 bar. The data points ($+$[10], $\bullet$[80]) mark prior experimentally determined melting points of ammonia-water mixtures. The points (X) mark the experimental temperatures (273 K, 298 K, 1 bar) and concentration (20.5 wt.% ammonia) of the samples studied herein.

ocean were influenced by astronomical studies of the environment of Young Stellar Objects that measured ammonia ice abundances of up to ~15% with respect to water-ice[22]. Despite the predictions of extensive reservoirs of ammonia hydrate condensate in the outer solar system[11,23], sample analysis and spectroscopic surveys suggest that ammonia seems to have survived only locally where it formed in stoichiometric excess[10,24]. Chondritic meteorites and asteroids sample relatively primitive, unmelted condensate from the solar nebula[25], and they exhibit significant variability in ammonia abundance. The carbonaceous meteorites of the Renazzo-type contain ammonia of up to 10% by mass of the insoluble organic material fraction, values far in excess of ammonia abundances in other carbonaceous meteorites[26] and the Ryugu asteroid[4]. Photometric and spectroscopic surveys of cometary nuclei suggest their ammonia concentrations are in the range of 0.2%–1.4% with respect to water abundance[27]. The difference in the predictions of chemical models and the results of sample analysis has been explained by the sequestration of ammonia by other volatiles and salts such as carbon dioxide, formaldehyde and magnesium sulphate to form ammonium salts[10]. Visible-Infrared mapping of the surface of Ceres, a dwarf planet in the main asteroid belt, finds the signature of ammonia salts that are consistent with ammonia sequestration[28]. Spectroscopic studies of Titan have proved difficult to interpret because its thick haze and cloudy atmosphere block the transmission of many visible and IR wavelengths, leaving only limited windows in which spectral observations can be made[29]. Few compounds have been identified beyond contaminated water-ice[29]. Infrared spectroscopy from the Mauna Loa observatory suggests the presence of ammonia hydrate, or flash frozen ammonia water, on the surface of the Uranian moons Miranda[30,31] and Ariel[32]. Cassini's Ion and Neutral Mass Spectrometer flew through Enceladus's ice-water plume in 2008 and measured a volume mixing ratio of 90% water and 0.82 +/− 0.02% ammonia[33] that the authors suggest is circumstantial support for the presence of ammonia in Titan.

Figure 1b illustrates that at 1 bar, ammonia-water crystallises at low temperature to form three hydrates: ammonia monohydrate ($H_2O:NH_3$), ammonia dihydrate ($2H_2O:NH_3$) and ammonia hemihydrate ($H_2O:2NH_3$). To understand the geophysics of the cores of the ice-giants, which models suggest are rich in ammonia, methane and water[34], X-ray, and neutron diffraction studies have been performed on solid state ammonia hydrates under high pressures and a range of temperatures[35–37]. In 2023, the National Academy of Sciences prioritised the Uranus Orbiter and Probe as the highest-priority new Flagship mission for initiation in the decade

2023–2032[38], hence we anticipate that studies of the molecular structure of water-ammonia-methane systems, will receive increasing attention. In this work, we explore the intermolecular structure of aqueous ammonia at temperatures and pressures that are relevant to a cooling impact melt on the surface of Titan that is enriched in ammonia[2].

Neutron diffraction studies of simplified mimetic solutions, water-ices and clathrate hydrates have proven to be a powerful tool in exploring extreme environments on Earth and in the icy worlds of the solar system[39–43]. The ammonia concentration of Titan's sub-surface ocean is unknown, but a 14 wt.% ammonia-water layer beneath an ice-water (Ih) shell of 75 km thickness has been modelled[21]. Cynn et al. modelled a sub-surface Titan ammonia-water ocean containing a mole fraction of 0.16 ammonia based on the cosmic abundance of ammonia[23]. More recently, Leitner and Lunine modelled a "maximum value of 5% ammonia" in Titan's ocean[44]. Subsequent local processes might alter ammonia concentration, for example, it has been suggested that progressive freezing of an ammonia-water liquid in a vertical ice fissure may result in the buoyant propagation of an ammonia-water cryolava approaching the eutectic composition (33 wt.%)[45]. In a cooling impact melt, it has been suggested that the ammonia concentration might be expected to accumulate as the liquid begins to freeze[46].

Narten used X-ray diffraction to study aqueous ammonia at 297 K in 1968[47] but could not derive any information about the mutual orientation of water and ammonia molecules. Neutron diffraction, owing to the large scattering length of hydrogen and its sensitivity to hydrogen/deuterium substitution, is much more capable of resolving this information. Recent neutron diffraction studies have focused on pure liquid ammonia[48], or the high-pressure ammonia-water-ices, ammonia monohydrate[49] and ammonia dihydrate[50]; we are not aware of any published neutron diffraction studies of aqueous ammonia-water. Ricci et al.[48] concluded in a study of liquid ammonia at 213 K, 0.121 MPa and 273 K, 0.483 MPa, that whilst ammonia molecules exhibited some local intermolecular correlations, they did not form an extended hydrogen bonded network. Thompson et al.[51] in another neutron diffraction study on liquid ammonia at 230 K, 1 bar, found evidence for a weak hydrogen bond at 2.4 Å.

Spectroscopic studies of water-ammonia mixtures in helium drops[52] and in aqueous solution[53], show evidence for an ammonia-water hydrogen bond asymmetry and this result has been replicated by classical and ab initio molecular dynamics (MD) studies[53–56]. Ammonia's nitrogen atom has been found to accept a medium-strong hydrogen bond from water, but if ammonia donates hydrogen bonds to water, they have been described as

ultra-weak[53]. Ab initio molecular dynamics (MD) studies suggest that the weak $H_A-O_W$ interaction (atom label conventions shown in Fig. 1) is due to an energetic mismatch in orbital energies that prevents effective charge transfer[53]. Recent ab initio molecular dynamics simulations of ammonia-water solutions at 295 K, and at 90:10, 75:25, 50:50, 25:75, water:ammonia molar ratios, emphasise the importance of the proton affinity of the nitrogen atom of ammonia in aqueous solution, and suggest that at higher ammonia concentrations, the $N_A-H_W$ interaction is weakened[54]. Neutron powder diffraction data suggest that the crystal structures of the high-pressure ices, ammonia monohydrate and ammonia dihydrate, feature only the $N_A-H_W$ hydrogen bond[57].

To better understand the effect of ammonia, if it is present, on prebiotic chemistry at Dragonfly's landing site, the Selk impact crater, and ammonia-rich fluids in other ocean worlds, there is a need to complete experimental intermolecular structural studies of these fluids at conditions relevant to

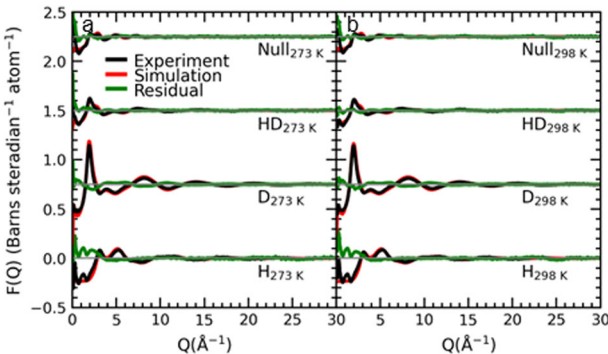

**Fig. 2 | The experimental and simulated $F(\mathbf{Q})$. a, b** describe the fits of simulated $F(\mathbf{Q})$ from models of the solution system to experimental diffraction measurements of aqueous samples of 20.5 wt.% ammonia solutions at 273 and 298 K, respectively. The fits are displaced vertically by 0.75 units to aid comparison. The $F(\mathbf{Q})$ is reported for 4 isotopically substituted samples where H and D represent the light and heavy isotopes of hydrogen, and HD and Null represent 50:50 and 64:36 H:D isotopic fractions. The description 'Null' pertains to the hydrogen isotopic composition of water that exhibits no overall coherent scattering effects from the hydrogen nuclei. The Residual describes the difference between the experimental and simulated $F(\mathbf{Q})$.

their host environments. This study uses neutron diffraction with isotopic substitution (NDIS), and computer modelling of the diffraction datasets using Empirical Potential Structure Refinement (EPSR)[58], to study a 20.5 wt.% (21.4 mol%) solution of ammonia in water at 1 bar at 273 K, and at 298 K utilising the Near and InterMediate Range Order Diffractometer (NIMROD)[59], at the ISIS Neutron and Muon Source, Didcot, UK. In Fig. 1b, we illustrate the experimental conditions for the current study alongside the experimental melting points of ammonia-water mixtures. The results are compared to previous NIMROD measurements of pure water at 298 K, 1 bar[60]. A 20.5 wt.% ammonia solution is broadly similar to conditions expected at a cooling Titan impact melt, or cryolava, above the eutectic, and concentrated enough to clearly observe the perturbing effects of the solute on the water structure with neutron diffraction. The experimental conditions are also similar to the experimental conditions of Titan-tholin hydrolyses in aqueous ammonia, Table S1[2,46,61–63].

We show that in solution, ammonia molecules interact very weakly with each other, whilst exhibiting an asymmetry in the strength of their hydrogen bond interactions with water molecules; ammonia donates weak hydrogen bonds to water and strongly accepts hydrogen bonds from water, verifying the predictions of recent MD simulation studies[54,56]. In this work, we show how the strong water to ammonia hydrogen bond ($H_W-N_A$) is co-operatively translated into ammonia's hydration shell, where we observe the formation of ice-like motifs.

## Results

In Fig. 2a, b, we report the experimentally measured neutron structure factors, $F(\mathbf{Q})$, following correction for inelasticity, attenuation, and multiple scattering effects using Gudrun software[64], for aqueous 20.5 wt.% ammonia at 273 K and 298 K, respectively. The $F(\mathbf{Q})$ describes the summation of all interatomic pair correlations $S_{ab}(\mathbf{Q})$ weighted by their atomic fractions and neutron scattering lengths (Eq. 1) that were used to refine simulations in EPSR of the solution structure, based on the intramolecular geometries and atomic densities. Figure 2 demonstrates that the $F(\mathbf{Q})$ simulations provide a good fit to the experimental $F(\mathbf{Q})$.

In Fig. 3a, we compare the $g_{O_W-H_W}(r)$, and in Fig. 3b, the $g_{O_W-O_W}(r)$ radial distribution functions relating to the ammonia-water samples (273 K, 298 K) and pure water (298 K). These describe the local density of a particular atom type relative to a central atom type as a function of distance, normalised to the bulk density of the experimental sample. The series of

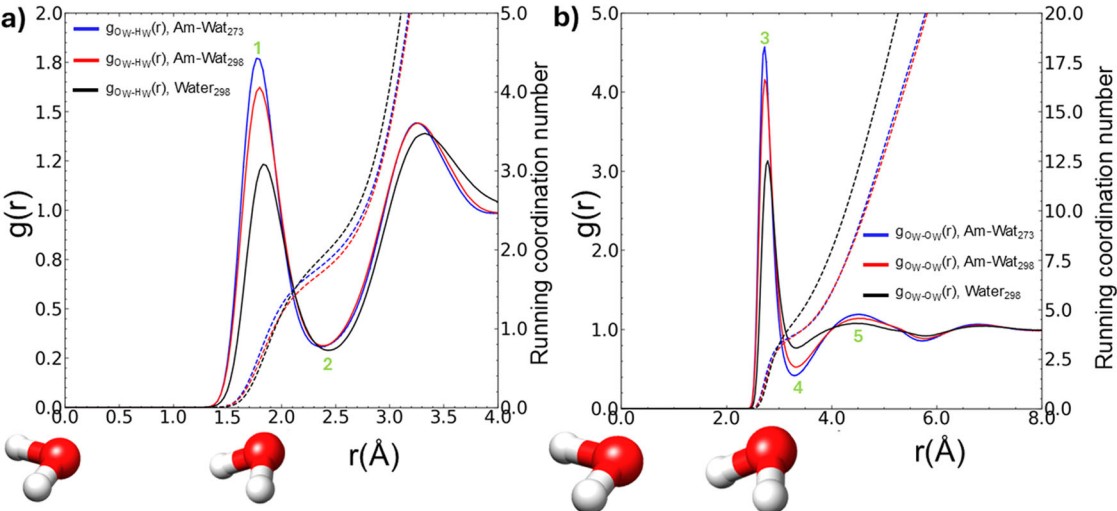

**Fig. 3 | The atomic pair distributions, $g_{O_W-H_W}(r)$ and $g_{O_W-O_W}(r)$, in aqueous ammonia and pure water.** In **a, b** we compare the $g_{O_W-H_W}(r)$ and $g_{O_W-O_W}(r)$ in aqueous ammonia (273 K, 298 K) and in pure water (298 K), respectively, all at 1 bar. The dashed lines in (**a**) describe the co-ordination of a central $O_W$ by surrounding $H_W$ atoms at $r$, and the dashed lines in (**b**) describe the co-ordination of a central $O_W$ by surrounding $O_W$

atoms at $r$. Inset: An oxygen atom of water is placed at the origin of the coordinate system, and alongside it is a 2nd water molecule as a guide to features in the $g(r)$. 1,2 and 3,4 mark the modal and maximum bond lengths of the $O_W-H_W$ and $O_W-O_W$ interactions, respectively. 3 marks the location of the 1st hydration shell of the water network, and 5 marks the location of the 2nd hydration shell.

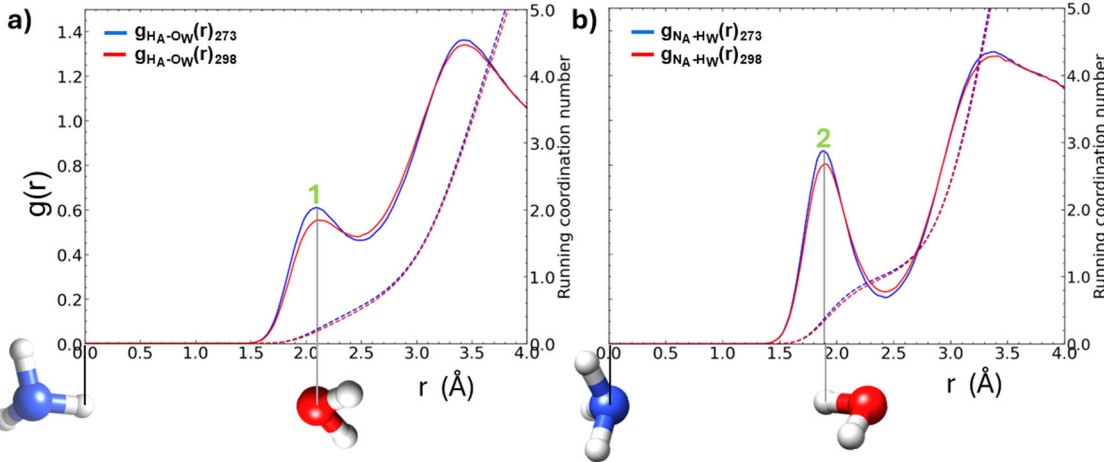

**Fig. 4 | The atomic pair distributions, $g_{H_A-O_W}(r)$ and $g_{N_AH_W}(r)$ in aqueous ammonia.** In (**a**) we compare the $g_{H_A-O_W}(r)$ at 273 K, 298 K, 1 bar, and in (**b**) we compare the $g_{N_AH_W}(r)$ at 273 K, 298 K, 1 marks the modal $H_A-O_W$ hydrogen bond distance (2.10 Å) at 273 and 298 K. The dashed blue and red lines mark the co-ordination numbers of $O_W$ around $N_A$ (**a**) and $H_W$ around $N_A$ (**b**) at 273 and 298 K. 2 marks the modal $N_A-H_W$ hydrogen-bond distance (1.89 Å) at 273 K, 298 K.

**Fig. 5 | The atomic pair distributions, $g_{N_A-H_A}(r)$, $g_{N_A-N_A}(r)$ and $g_{H_AH_A}(r)$ in aqueous ammonia.** The limited interactions between ammonia molecules at 273 K, 298 K are described by **a** $g_{N_A-H_A}(r)$, **b** $g_{N_A-N_A}(r)$, and **c** $g_{H_A-H_A}(r)$.

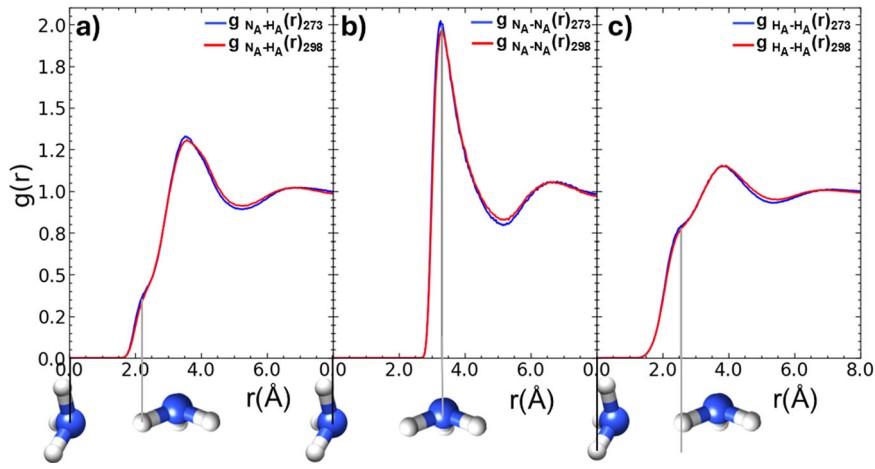

peaks and troughs in the $g(r)$ describe atomic co-ordination shells that decay to unity as the distance becomes sufficiently large that bulk behaviour is approached.

The first maxima and minima of the $g(r)$ describe the modal interatomic distance and the maximum extent of the co-ordination shell of each atom-pair interaction, respectively. The peaks in the $g(r)$ are more prominent, i.e., the first maxima and first minima are higher and lower, respectively, in the presence of ammonia. The width of the first peaks at half maximum is also narrower in the presence of ammonia and at lower temperatures. These effects demonstrate that the variance in the modal distances between water atoms is reduced by ammonia addition, an effect partially counteracted by increasing temperature. Table S2 summarises the observed peak positions and co-ordination numbers for the $g(r)$ illustrated in Fig. 3.

The co-ordination of a central water atom, $O_W$, by $H_W$ atoms was measured to be 1.81 and 1.73 in the aqueous ammonia samples at 273 K and 298 K, respectively, up to 2.40 Å, indicating that the water network is temperature sensitive, an effect observed in diffraction studies of super-cooled water (244 K)[65]. In contrast, in pure water, at 298 K, this co-ordination number was measured to be 1.94 (2.40 Å), comparable to other measurements in the literature[65,66]. Ammonia addition leads to a reduction in the co-ordination of $O_W$ by $H_W$ by 11% from 1.98 to 1.73, yet the $H_W$ atom fraction is diluted by 27% (the change in atomic density is negligible)

indicating a resilience in the remnant water network in aqueous ammonia. In Fig. 3b, we also compare the $g_{O_W-O_W}(r)$ in pure water to aqueous ammonia. The 2nd peak in the $g(r)$ (~4.5 Å) marks the location of the 2nd hydration shell[67]. The $g(r)$ at the 2nd peak was 1.14 in aqueous ammonia at 298 K, compared to 1.08 in pure water at 298 K. The result that ammonia addition is associated with a more defined 2nd hydration shell of water is consistent with an enhanced tetrahedral structure in the water network.

In Fig. 4a, we illustrate the $g_{H_A-O_W}(r)$, and in Fig. 4b, the $g_{N_A-H_W}(r)$ at 273 K, 298 K. The height and location ($r$) of the first peaks indicate a strong asymmetry in ammonia-water hydrogen bonding; the $N_A-H_W$ interaction is much stronger than the $H_A-O_W$ interaction. In the aqueous ammonia sample at 273 K, we find that on average, a central nitrogen atom of ammonia is coordinated by 1.02 hydrogen atoms of water inside the maximum hydrogen bond distance of 2.43 Å. In contrast, a central hydrogen atom of ammonia is coordinated by 0.54 oxygen atoms of water inside the maximum bond distance of 2.49 Å at 273 K. The first maxima of the $g_{H_A-O_W}(r)$, marks the modal distance between the atoms (Fig. 4a) of 2.10 Å. In comparison, the $N_A-H_W$ modal distance is 1.89 Å at both 273 K and 298 K (Fig. S2b).

To assess the extent of ammonia-ammonia hydrogen bonding, Fig. 5 illustrates the $g_{N_A-H_A}(r)$, $g_{N_A-N_A}(r)$, and $g_{H_A-H_A}(r)$, at 273 K, 298 K. The shoulder in the $g_{N_A-H_A}(r)$, at 2.2 Å evidences a very weak interaction. The weak features at ~6.6 Å in the $g_{N_A-N_A}(r)$, and at ~2.4 Å in the $g_{H_A-H_A}(r)$,

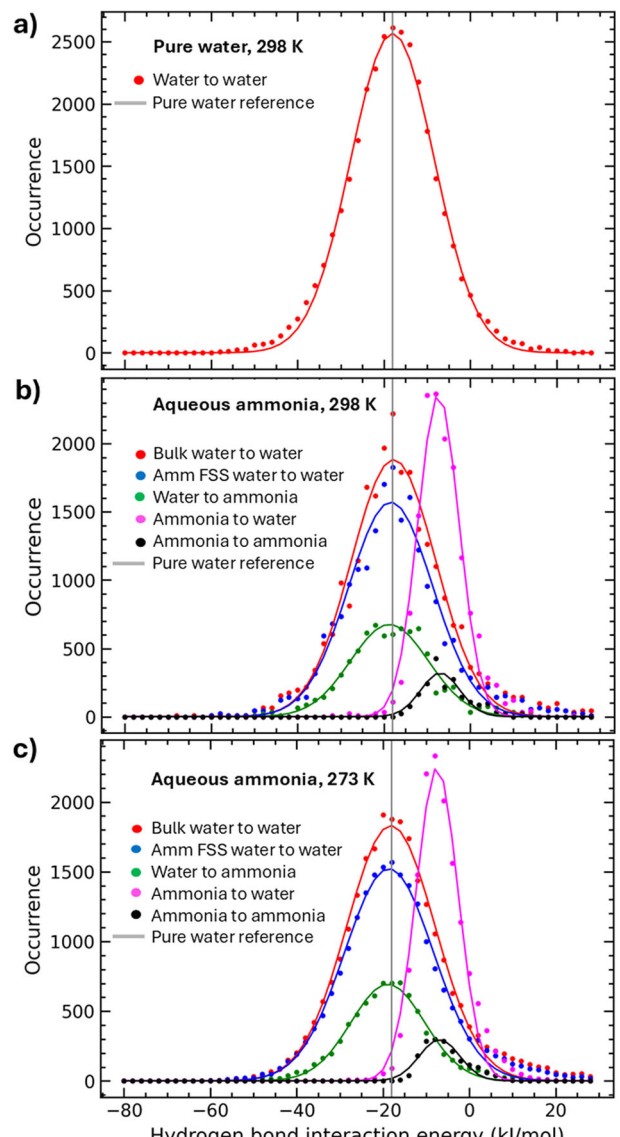

**Fig. 6 | The hydrogen-bond enthalpies of pure water and aqueous ammonia.** The hydrogen bond enthalpies of pure water at 298 K (**a**), and bulk water, water in the first solvation shell (FSS) of ammonia, water to ammonia, ammonia to water, and ammonia to ammonia at 298 K (**b**) and 273 K (**c**). 'to' describes the direction of the hydrogen bond. The description, 'Ammonia FSS Water', describes hydrogen bonds between water molecules, where the first water molecule is hydrogen-bonded to an ammonia molecule. The data is drawn from the atomic co-ordinates of molecular trajectories derived from EPSR simulations combined with a script to quantify the interaction energy[60].

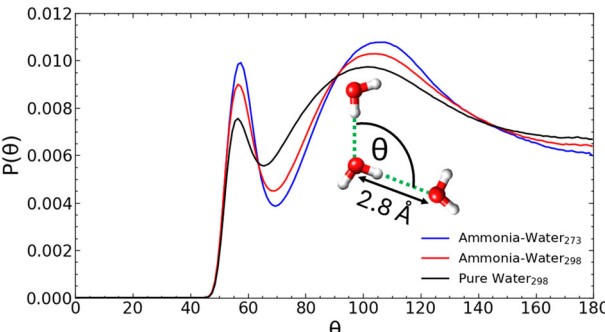

**Fig. 7 | The tetrahedrality of the water network in pure water and aqueous ammonia.** The angular distribution of $\angle O_W$-$O_W$-$O_W$ atoms in the ammonia-water samples at 273 K, 298 K and in pure water at 298 K. We report here only the angular distribution of $O_W$ triplets where the $O_W$-$O_W$ distance is $\leq 3.3$ Å. Inset: a sketch of the tetrahedral hydrogen bonding pattern particularly favoured by water molecules in aqueous ammonia.

The data points correspond to hydrogen bonds identified in the configurational ensembles of EPSR. The lines are Gaussian fits from which the mean interaction energies were derived (Table S3).

The results (Table S3) indicate that the water to ammonia hydrogen bond ($H_W$-$N_A$, $-18.74$ kJ mol$^{-1}$, 298 K) is the strongest in solution, and that in donating a hydrogen bond to ammonia, the hydration shell of water within the first solvation shell of ammonia is co-operatively strengthened (the water-water enthalpy is $-18.42$ kJ mol$^{-1}$, compared to $-18.01$ kJ mol$^{-1}$ in pure water at 298 K) indicating that co-operative hydrogen bonding[9,68] is the underlying mechanism for enhanced tetrahedral organisation in the remnant water network. Consistent with the reported $g(r)$, we observe a large hydrogen bond asymmetry in aqueous ammonia, where hydrogen bonds donated to ammonia by water molecules are much more enthalpically stable than those donated to water by ammonia. This weak hydrogen bond donating propensity of ammonia is highlighted by very weak and infrequently occurring ammonia-ammonia hydrogen bonds ($-6.94$ kJ mol$^{-1}$ at 298 K).

The tetrahedral structure of liquid water is manifest in the $\angle O_W$-$O_W$-$O_W$ triplet bond angle distribution of hydrogen bonded water molecules. In Fig. 7, we compare this angular distribution in pure water to that in aqueous solutions of ammonia. In pure water, the distribution exhibits a broad peak at 101.5°. In aqueous ammonia (298 K), the peak is at 104.5°, and a greater proportion of the network adopts a tetrahedral (~109.5°) bond angle. There is also an enhanced probability for water molecules to coordinate with a ~ 60° bond angle, a tendency apparent in pure water. The 60° bonding arrangement results from the intrusion of a water molecule from the 2nd hydration shell into the first hydration shell of a central water molecule, so-called 'interstitial' water molecules.

In Fig. 8, we compare the topology of the water network in pure water at 298 K (1 bar) to 20.5 wt.% ammonia solutions at 273 K, 298 K (1 bar). To show this, we define a hydrogen bonded water linkage where the $O_W$-$O_W$ distance $\leq 3.3$ Å and the included angle $\leq 30°$ (Fig. S1). In panel (a), we report that ammonia addition is associated with an increase in the size of the ringed water complexes defined by the shortest path. In panel Fig. 8b, we report the size distribution, by the shortest path, of linear chains of such hydrogen-bonded water molecules. Ammonia addition enhances the extent of large linear chains of water molecules. In contrast, in panel Fig. 8c, we show that ammonia addition reduces the size of water clusters, where a water cluster is an assemblage of water molecules where each oxygen atom is $\leq 3.3$ Å from a neighbouring oxygen atom. Figure S2 demonstrates that ammonia molecules did not form a hydrogen bonded network in the aqueous ammonia models.

To evidence the spatial relationship of water-ammonia molecules, the spatial density distribution of water and ammonia molecules around a central water molecule is presented in Fig. 9.

suggest some local correlation of $N_A$-$H_A$ atoms, but that ammonia clustering is very limited. There is negligible observed temperature dependency in the ammonia-ammonia $g(r)$.

Figure 6 describes the hydrogen-bond enthalpies, using a routine previously devised to study water's hydrogen-bond network[60]. The routine, outlined in the methods, identifies hydrogen bonded species by reference to (i) their atomic co-ordinates in the EPSR simulation, (ii) geometric criteria for the hydrogen bond, and the measured maximum hydrogen bond lengths. Figure 6 shows the hydrogen bond enthalpies for pure water at 298 K (a), bulk water in aqueous ammonia (here defined as any water molecule where the $O_W$ atom does not lie within 3.6 Å of an $N_A$ atom), water in the first solvation shell (FSS) of ammonia, water to ammonia ($H_W$-$N_A$), ammonia to water ($H_A$-$O_W$), and ammonia to ammonia ($H_A$–$N_A$) at 298 K (b) and at 273 K (c) where 'to' refers to the direction of the hydrogen bond.

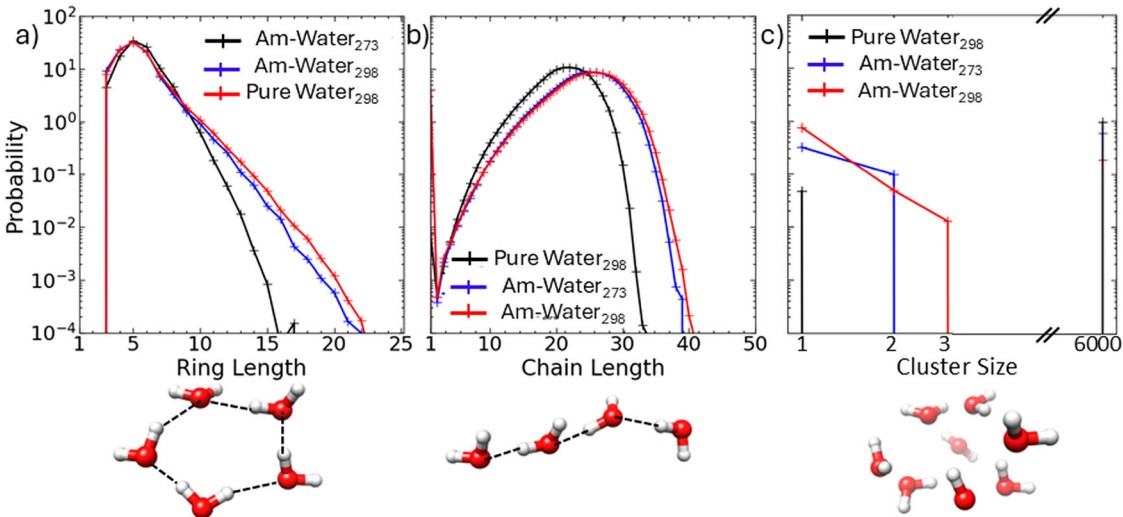

**Fig. 8 | Topological characteristics of pure water and aqueous ammonia.** The topological change in the water network is rendered by ammonia addition. In (**a**), we report the size distribution of rings of hydrogen bonded water molecules by the shortest path. In (**b**), we illustrate the size distribution of linear chains of hydrogen bonded water molecules that fulfil a shortest path criterion. The linear and ring hydrogen bond complexes are defined such that the $O_W$-$O_W$ distance is ≤3.3 Å, and the included bond angle ≤30°. In (**c**) the cluster size distribution is illustrated, where a cluster is more loosely defined to be those water molecules whose nearest neighbour fits criteria $O_W$-$O_W$ ≤ 3.3 Å. Inset: Examples of ringed and linear hydrogen bonded water complexes, and a cluster of water molecules.

We place a water molecule at the origin, define an axis of symmetry through the water molecule, and using an ensemble of atomic co-ordinates from EPSR, we report the spatial distribution of $O_W$, and $N_A$ atoms within 3.5 Å of the central $O_W$ atom. The distribution of water molecules in pure water (Fig. 9a) provides a benchmark. In Fig. 9b, the asymmetry of water-ammonia hydrogen bonding is apparent; the $N_A$, and $O_W$ occupy similar regions around the central $H_W$ atoms, but around the central $O_W$ atom, the density of $N_A$ atoms is diffuse. If we compare the distribution of $O_W$ around the central $O_W$ in the pure water model in Fig. 9a to the ammonia-water model in Fig. 9b, $O_W$ atoms are more spatially confined by ammonia addition, as some ammonia molecules occupy interstitial space in water's tetrahedral motif. In Fig. 10, we show the spatial distribution of water and ammonia molecules around an ammonia molecule and again find ammonia molecules occupying interstitial space in the ammonia-water network. This arrangement is in accordance with the pair-wise interaction energies (Table S3). Supplementary movies 1 and 2 illustrate the spatial density distributions presented in Figs. 9b and 10b, respectively, rotated through 360° around the vertical axis.

## Discussion

This study shows that ammonia molecules exhibit some weak electrostatic correlations between their nitrogen and hydrogen atoms, but that ammonia molecules do not form a hydrogen bonded network (Figs. 5 and 6, Fig. S2). In Table S3, we report that the $N_A$-$H_A$ hydrogen-bond enthalpy was the weakest in solution (−6.9 kJ mol⁻¹). This weak self-interaction of ammonia molecules is in accordance with the low-temperature neutron diffraction study of pure liquid ammonia by Ricci et al.[48].

Spectroscopic studies of water-ammonia mixtures in helium drops[52] and MD studies of aqueous ammonia[53–56], show evidence for an ammonia-water hydrogen bond asymmetry that we also report in Figs. 4 and 6. Table S3, reports our energetic analysis of hydrogen bonds in solution and shows hydrogen bond enthalpies of −7.48 kJ mol⁻¹ and −18.89 kJ mol⁻¹ for the $H_A$-$O_W$ and $N_A$-$H_W$ interactions, respectively. This result accords with the height and locations of the first peaks in the $g_{H_A-O_W}(r)$ and $g_{N_A-H_W}(r)$ that describe these hydrogen bond length distributions. We found that ammonia was twice as likely to bond to water through its nitrogen atom as opposed to its hydrogen atoms.

It was anticipated that the ammonia molecule would exhibit an asymmetry in the strength of its hydrogen bond interaction with water, and that this might curtail the extent of the hydrogen bonded water network. Our results confirmed this to be so, but unexpectedly we found that ammonia addition strengthens the enthalpy of the remnant water network, and both enhances the tetrahedral organisation of the water network, and the extent of large linear and ringed hydrogen bonded clusters. The result that ammonia addition leads to ice-like-ness in its hydration shell is reminiscent of the predictions of Frank and Evans[5], who had suggested that when a non-polar molecule dissolves in water, it passively modifies the surrounding molecular layers of water structure in the direction of greater crystallinity. In contrast, in aqueous 20.5 wt.% ammonia, at 273 K, our enthalpic analysis (Fig. 6 and Table S3) suggests that the strong $N_A$-$H_W$ interaction (−18.89 kJ mol⁻¹) influences the water network as its electronic effects are co-operatively translated through the ammonia hydration shell ($H_W$-$O_W$, 18.60 kJ mol⁻¹) into the bulk water network ($H_W$-$O_W$, 18.16 kJ mol⁻¹). That liquid water's hydrogen bond environment can rapidly rearrange in response to solute addition is well known[9], and has been demonstrated by MD studies in aqueous ammonia[56]. In this work, we have highlighted the role of the $N_A$-$H_W$ interaction in enhancing the tetrahedrality of water molecules. The ice-like-ness of the water network in aqueous ammonia is made explicit in this work for the first time, but MD studies have also highlighted the role of cooperative hydrogen bonding in modulating both networks. They also link the strength of the $N_A$-$H_W$ bond to tetrahedral co-ordination of water molecules in ammonia's hydration shell[56], but in reverse, emphasising that it is the tetrahedrality of the hydrogen-bonded water complex in ammonia's hydration shell that strengthens the $N_A$-$H_W$ bond. They also suggest that as ammonia concentrations are increased, the $N_A$-$H_W$ hydrogen bond might weaken as electron pairs of the water molecules would be shared by an increasing number of $H_A$ atoms[54]. The MD studies also highlighted the significance of the proton affinity of the nitrogen atoms of ammonia in aqueous ammonia solutions[54] and have quantitatively testified that the $NH_3$ species are more prone to accept protons than $H_2O$ molecules[56]. The MD results are compatible with our results and underline the role of co-operative hydrogen bonding in modulating hydrogen-bond networks. Steric packing of non-bonded ammonia molecules, in occupying the interstices of the water hydrogen bond network, may also play a role in ordering the water network.

The organisation of water by ammonia may have implications for the hydrophobic interaction, a significant force in molecular biology[69]. Historically, Kauzmann linked this interaction to enhanced structure in water around non-polar solutes like methane, or non-polar organic atoms[6]. Over the years, support for this theory waned[69], as direct structural evidence for

**Fig. 9 | The spatial distribution of water and ammonia molecules around a central water molecule.** The spatial density distribution of $N_A$ atoms of ammonia (blue) and $O_W$ atoms of water molecules (red) around a central water molecule in **a** pure water (298 K), **b** ammonia-water (273 K). The surface contours contain the most probable (top 70%) locations for finding water atoms ($O_W$) and ammonia atoms ($N_A$) inside 3.5 Å.

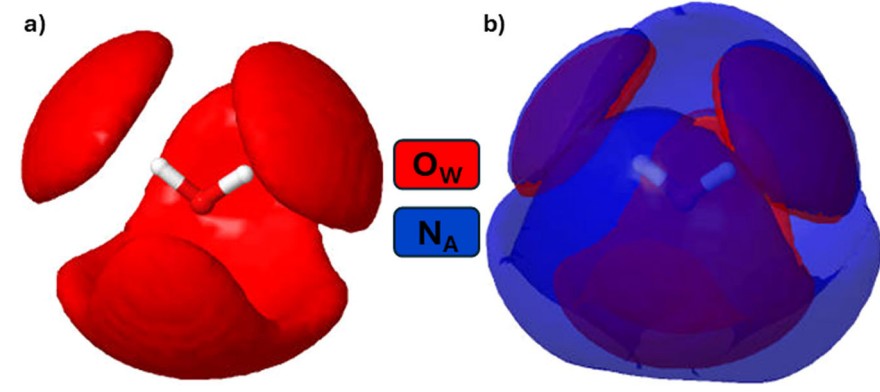

**Fig. 10 | The spatial distribution of water and ammonia molecules around a central ammonia molecule.** The spatial density distribution of $N_A$ atoms of ammonia (blue) and $O_W$ atoms of water molecules (red) around a central ammonia molecule in ammonia-water (273 K). Panel **a**, **b** offer different perspectives of the same distribution: **a** is from the 'top' down, and **b** from the 'side'. The surface contours contain the most probable (top 20%) locations for finding water atoms ($O_W$) or ammonia atoms ($N_A$) inside 4.0 Å. Note that ammonia distribution is much more diffuse than water and that water molecules prefer to sit around the $N_A$ atom rather than the $H_A$ atom. The unmistakable impression is that ammonia sits in the interstices of water's energetic minimal landscape.

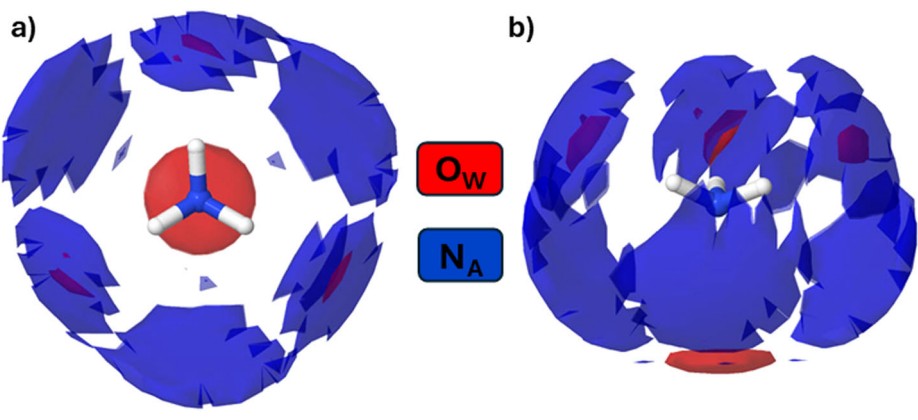

such quasi-crystalline states around hydrophobic atoms has been lacking[70]. Some insight into the possible structure of Frank-Evans icebergs was provided by the structure of clathrate hydrates, which are open crystalline networks of hydrogen bonded water that host small guest molecules such as methane. In aqueous ammonia, our experiments have shown that the nitrogen atom of ammonia organises a hydrogen bond network of water around it and increases water tetrahedrality in the remaining network. This leads us to speculate on the formation of clathrate-like structures around hydrophobic solutes in aqueous ammonia as compared to pure water. The aggregation of hydrophobic solutes in aqueous ammonia might incur reduced entropic advantage, resulting in reduced association of hydrophobic solutes in aqueous ammonia. The result that ammonia perturbs the hydrogen bond structure of liquid water also leads to a hypothesis that ammonia might similarly perturb clathrate structure, and this result is compatible with high-pressure experimental work on a 10 wt.% ammonia solution that has been shown to lower the dissociation temperature of methane clathrates by 14–25 K at pressures above 5 MPa[71]. This is an interesting result because it implies that the ammonia molecule might generally disrupt the structure of liquid water around hydrophobes and thus reduce the extent of the hydrophobic effect around prebiotic molecules, with implications for the self-assembly of prebiotic molecules in ammonia-rich ocean worlds. It remains to be seen just how prevalent aqueous ammonia is in the solar system's icy moons, but the results of this experiment suggest that the ammonia molecule, in significantly organising the surrounding water network, has structural effects on water that may have consequences for the assembly of the building blocks of Life.

## Methods
### Sample preparation
A 20 wt.% ammonia solution was purchased from Fisher Scientific, Loughborough, and 25 wt% deuteriated ammonium-d4 deuteroxide in $D_2O$

was purchased from Sigma Aldrich. Both samples were >99% pure and used without further purification. Stock solutions (1 mole ammonia: 3.665 moles water) of fully protiated and fully deuteriated ammonia solutions were created by diluting the deuteriated sample in $D_2O$. Appropriate amounts of each stock solution were measured by an analytical mass balance to create ~1.5 ml aliquots of four aliquots of isotopically substituted samples of composition: $H_2O$-$NH_3$, $D_2O$-$ND_3$, $HDO$-$NH_{1.5}D_{1.5}$, $H_{1.28}D_{0.72}$-$NH_{1.92}D_{1.08}$ at a mole ratio of 1:3.665, ammonia:water. At these mole ratios, the $N_A$ atom is the smallest atom fraction (~7%), which was considered sufficient to allow for its structural signal to be detected (Eq. 1).

Approximately 1.5 ml of each sample was transferred by syringe to labelled flat plate geometry, null-scattering alloy ($Ti_{0.68}Zr_{0.32}$) cans of 1 mm path length and 1 mm wall thickness. The cans were sealed with polytetrafluoroethylene (PTFE) and mounted in a sample changer, with temperature-controlled (273 K, 298 K) by a water bath recirculating through the sample can mounting frame.

### The neutron diffraction experiment
The sample changer was loaded into the Near and InterMediate Range Order Diffractometer (NIMROD)[59], at the ISIS neutron and muon source, the sample chamber evacuated, and the temperature of the sample chamber was reduced to 273 K. Each sample was in the beam for up to 4 h over a period of around 24 h with neutron scattering detected by banks of ZnS-based scintillation detectors. To calibrate the instrument, a plate of null-scattering VNb alloy of 3 mm thickness, with known scattering characteristics, was also placed in the beam for 2 h, under identical experimental conditions. Empty TiZr null-scattering alloy cans were measured in the beam for up to 2 h, and neutron diffraction with an empty sample changer was also measured. The measurements were then repeated at 298 K. On completion of the experiment, the scattering from the empty cans, the sample background,

and inelastic scattering from the hydrogen nuclei was removed using the data reduction package, Gudrun[64].

This experiment used a technique described as Neutron Diffraction with Isotopic Substitution (NDIS)[72]. NDIS exploits the contrast in the scattering power of different isotopologues that generate a unique differential scattering cross-section from isotopically substituted samples while preserving molecular structure. The coherent scattering length of hydrogen is $-3.74$ fm[73], and its isotope deuterium exhibits a large contrast in scattering length at 6.67 fm. The scattering effect is described by the measured total structure factors ($F(\mathbf{Q})$) that are the weighted sums of individual partial structure factors ($S_{\alpha\beta}(\mathbf{Q})$) of each type of scattering centre pair (atom–atom correlations ($\alpha,\beta$) Eq. 1).

$$F(Q) = \sum_{\alpha,\beta \geq \alpha} \left(2 - \delta_{\alpha,\beta}\right) C_\alpha C_\beta b_\alpha b_\beta \left(S_{\alpha\beta}(Q) - 1\right), \qquad (1)$$

where $c$ is the fractional atomic concentration, $b$ is the nuclear scattering length, $\delta$, the Kronecker delta, and $\mathbf{Q}$ is the scattering vector (Eq. 2) with units of Å$^{-1}$.

$$Q = \frac{4\pi \sin(\theta)}{\lambda} \qquad (2)$$

where $2\theta$ is the scattering angle, and $\lambda$ is the neutron wavelength.

We selected mixtures of fully protiated and deuteriated samples, since all hydrogen atoms are exchangeable in both water and ammonia molecules. In making these isotopic substitutions, we obtain composite structure factors $S_{XX}(\mathbf{Q})$, weighted by their nuclides' scattering lengths and concentrations. In aqueous ammonia, the measured $F(\mathbf{Q})$ is dominated by the hydrogen-hydrogen $S_{HH}$ partial structure factor. To isolate information regarding the correlation of nitrogen and oxygen atoms, we prepared a null-scattering hydrogen sample whose isotopic composition (Eq. 1) nullifies the scattering due to hydrogen and deuterium nuclides, emphasising the contribution to the scattering pattern from the weighted $S_{NN}(\mathbf{Q})$, $S_{OO}(\mathbf{Q})$ and $S_{NO}(\mathbf{Q})$ structure factors.

The partial structure factor ($S_{\alpha,\beta}(\mathbf{Q})$) is related to the radial distribution function, $g_{\alpha\beta}(r)$, (the local density of atom $\beta$ around atom $\alpha$ as a function of their interatomic separation $r$, normalised by the bulk number density of $\beta$) of the atom pair: $\alpha,\beta$ by a Fourier transform (Eq. 3),

$$S_{\alpha\beta}(Q) = 1 + 4\pi\rho \int_0^\infty r^2 \left(g_{\alpha\beta}(r) - 1\right) \frac{\sin Qr}{Qr} dr, \qquad (3)$$

$$n_\alpha^\beta(r) = 4\pi\rho c_\beta \int_{r_{min}}^{r_{max}} r^2 g_{\alpha\beta}(r) dr. \qquad (4)$$

where $\rho$ is the experimentally determined atom number density of the sample. Integration of the $g(r)$ over the integration limits, $r_{min}$ and $r_{max}$ gives the co-ordination number, the average number of $\beta$ atoms around a central $\alpha$ in a radial shell between $r_{min}$ to $r_{max}$ from atom $\alpha$ (Eq. 4).

### Empirical potential structure refinement
In a molecule containing $i$ different atomic species, the number ($N$) of different atom-pair correlations ($\alpha$-$\beta$) is given by Eq. 5,

$$N = \frac{i(i+1)}{2}. \qquad (5)$$

For pure water, $i = 2$, and $N = 3$ ($O_W$-$O_W$, $H_W$-$H_W$, $O_W$-$H_W$). In ammonia-water ($NH_3$-$H_2O$), there are 2 molecules in which we define 4 atom types ($H_W$, $O_W$, $N_A$, $H_A$). With $i = 4$, we would need at least 10 distinct isotopologues to experimentally separate the 10 unique $g_{\alpha\beta}(r)$. The problem of inverting the diffraction data to real-space data grows significantly with sample complexity, and Empirical Potential Structure Refinement (EPSR) is a widely used structural refinement modelling system that provides a

solution to this problem[58,74]. Within EPSR, we use a priori knowledge of the components to simulate molecular models of the solution systems with user-defined molecular geometries, Lennard-Jones potentials, and a cubic box dimension set to constrain the simulation to the experimentally determined densities (Table S4). A molecular simulation was set up in EPSR and refined against the experimental diffraction measurements. After randomising the molecules, and equilibrating the box so that it adopts the most stable configuration, the Monte Carlo[75] simulation proceeds to explore intermolecular and intramolecular configurations by iteratively testing randomised atomic movements (whole molecular translations, rotations and individual atomic movements) against the metropolis condition such that the move is accepted if the potential energy change is negative, and if it is positive, the probability of the move being accepted is $e^{-\Delta U/kT}$ where $kT$ is the Boltzmann thermal factor, and $U$ a combination of a Lennard Jones potential and Coulomb potential. After initial equilibration, an additional empirical potential (EP) is derived from the difference between the measured and simulated $F(\mathbf{Q})$ (based on atomic positions in the box) and included in the calculation of $U$. Over time, the residual difference between the measured and simulated $F(\mathbf{Q})$ is reduced (Fig. 2). In this way, the pair potentials are perturbed by EP terms, derived from the diffraction measurements, and the structure in the simulation box is refined so that it is consistent with the measured structure factor derived from the measured diffraction datasets. It is important to note that EPSR does not guarantee a single unique solution to the data, but a series of time-averaged states that are compatible with the measured structure factor and based on sensible starting potentials.

### Partial ionisation
Ammonia is a weak base that undergoes partial dissociation on addition to water: $NH_3 + H_2O \rightleftharpoons NH_4^+ + OH^-$. The base dissociation constant ($k_b$) is $1.77 \times 10^{-5}$ at RTP[76], and $1.37 \times 10^{-5}$ at STP[77]. The degree of ionisation is concentration dependent, and is described by Ostwald's Dilution Law[78] and Eq. 6,

$$k_b = \frac{\alpha^2}{1-\alpha} c_0, \qquad (6)$$

where $\alpha$ is the degree of dissociation, and $c_0$ is the molar concentration of ammonia added to the solution. A 20.5 wt.% ammonia solution has a density of 0.92 g cm$^{-3}$ (ref. 79) and a molarity of 11.07. Solving for $\alpha$ in Eq. 6, the percentage of ionisation in 20.5 wt.% aqueous ammonia at RTP is 0.13%. Given this numerical insignificance, ammonia ionisation has been ignored in the modelling of scattering from the aqueous ammonia samples.

### Quantifying hydrogen-bond interaction energy
In studying the interaction of the small organic molecule trimethylamine-N-Oxide with water molecules, Laurent et al.[60] developed a routine to evaluate hydrogen-bond interaction energies, and that routine was used in this experiment. The data for the routine was a molecular trajectory written after the EPSR simulation had been optimised under the empirical potential. 11 frames were selected from this trajectory, 50 iterations apart, to allow for a statistically significant sample of uncorrelated molecular configurations. Each frame was read iteratively to identify 6 hydrogen bond interactions of interest. This was achieved by identifying the central hydrogen-bond acceptor of interest and then reviewing neighbouring atoms of interest within cut-off distances defined by the maximum hydrogen-bond lengths determined from the first minima in the $g(r)$s (Table S5). In water, as an example, the hydrogen bonds were identified where the central $O_W$ atom and the neighbouring $O_W$ atom were within 3.3 Å and the central $O_W$ and the neighbouring $H_W$ were within 2.4 Å. Having identified the hydrogen bonds, we evaluated their interaction energy by summing their interatomic Lennard-Jones and Coulomb potentials using the reference potentials for each atom type. The interaction energies were derived from this dataset by Gaussian fits.

**Article**

## Data availability

Figs. S1-S3 and Table S1-S5 are available through a Supplementary Information file. The experimental diffraction data and the EPSR simulations can be viewed at the data repository managed by the University of Leeds: https://doi.org/10.5518/1692.

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

## Acknowledgements

The project was supported by a grant from the Engineering and Physical Sciences Research Council (EPSRC) (EP/P02288X/1) and a European Research Council Consolidator Fellowship/UKRI Frontier Research Fellowship for the MESONET project, UKRI EP/X023524/1 to L. Dougan. We acknowledge the ISIS Neutron and Muon Source (Science and Technology Facilities Council) for provision of beamtime on NIMROD (https://doi.org/10.5286/ISIS.E.RB2220489).

## Author contributions

Dougan and Nasralla conceptualised the research. Alderman and Laurent assisted Nasralla with the experiment. Nasralla analysed the data. All the authors wrote the paper.

## Competing interests

The authors declare no competing interests.

## Additional information

**Peer review information** *Communications Chemistry* thanks Giuseppe Cassone and the other anonymous reviewer(s) for their contributions to this work's peer review. A peer review file is available.

