## [Transparent Peer Review file · Communications Chemistry]

Solution Structure of Titan-Relevant Aqueous Ammonia by Neutron Diffraction

Corresponding Author: Professor Lorna Dougan

Version 0:

Reviewer comments:

Reviewer #1

(Remarks to the Author)

See attachment

Reviewer #2

(Remarks to the Author)

This study investigates the structural effects of ammonia in aqueous solutions, particularly in the context of Titan and other ammonia-rich ocean worlds. Using neutron diffraction with isotopic substitution combined with Empirical Potential Structure Refinement modeling, the authors examine a 20.5 wt.% ammonia-water solution at 273 K and 298 K under 1 bar pressure. Their findings confirm the asymmetry in ammonia's hydrogen bonding with water, revealing that ammonia is twice as likely to bond via its nitrogen atom rather than its hydrogen atoms. Additionally, the study demonstrates an enhanced tetrahedral ordering of water around ammonia.

The figures illustrating the radial distribution functions and spatial density distributions of the atoms are clear and well-presented. The combination of neutron diffraction and computational modeling provides a robust support for the study's conclusions. This research is highly relevant to understanding Titan's aqueous chemistry and the prebiotic potential of ammonia-rich environments, with implications for the upcoming Dragonfly mission and future explorations of the Uranian system. I recommend the publication in the *Journal of Communications Chemistry* after minor revisions. Please see below a list of suggestions/questions that the authors might consider to improve the manuscript.

L89-93: There is increasing evidence suggesting that comets are the building blocks of outer solar system objects. Observations of comet 67P/Churyumov-Gerasimenko by the Rosetta mission imply that the abundance of ammonia is likely around ~1-2% relative to water [Mumma, M.J. and S.B. Charnley (2011). *Annual Review of Astronomy and Astrophysics*, 49: p. 471-524], rather than the ~5-15% predicted by early chemical models. I think this should be discussed here.

L115-116: Why did you select a 20.5 wt% ammonia concentration? Does this concentration more accurately reflect the conditions expected in Titan's impact melt pools, or is it meant to represent an average of the concentrations you mentioned earlier (14wt% and 33wt%)? You mentioned that these conditions enable a study relevant to Titan, but could you clarify whether this specific concentration was chosen to better simulate the conditions of Titan's cryolava or for another reason?

How would ammonia behave as the temperature of the impact melt pool decreases at lower temperatures (just above the eutectic for example)?

How do your results scale to higher ammonia concentrations or higher pressures (800 MPa) as found at the bottom of Titan's ocean?

Given that the results suggest ammonia significantly organizes the surrounding water network, could these findings provide insights into ammonia's interaction with methane clathrates on Titan, potentially favoring partial dissociation rather than the incorporation of ammonia into the clathrate structure?

Minor edits:

L165: It should be "the presence," not "the e presence."

L253: It should be "Figure 8c," not "S8c."

L413: There seems to be a typo with "figure S9."

Reviewer #3

(Remarks to the Author)

Dougan and co-workers report an interesting experimental investigation on the water structure in aqueous ammonia solutions. In particular, they show that ammonia addition leads to the formation of ice-like motifs in ammonia's hydration shell. Overall, the reported findings represent a valuable contribution to the community, even though some previous computational investigations already predicted an ammonia-induced structuring of the local water environment leading to important consequences in the reactivity of these acid-base systems [JPCL 14, 7808 (2023)]. In the following, I'm going to list a series of suggestions that the Authors could consider before submitting a revised version of the article for eventual publication.

*) The local water structuring induced by the addition of ammonia was predicted and reported by a series of ab initio molecular dynamics simulations of water-ammonia mixtures, mainly with the aim of disclosing the role of local molecular clustering in the formation of NH_4^+ and OH^- species [JPCL 14, 7808 (2023)]. In this contribution, the Authors show how the proton-accepting capability held by the N atom is capable of polarizing the H_2O H-bond donor molecule and, consequently, its hydration shell, leading to more structured local configurations. Although this represented only a computational prediction, due to the adherence of the results, it would be worth discussing these predictions also in the current investigation where, however, such a work is not mentioned.

*) Title, abstract, introduction, and the conclusion paragraph link the reported findings with aqueous ammonia solutions present in Titan. However, the remainder part of the article (i.e., the results paragraph) doesn't mention any plausible connections with what we know about ammonia aqueous solutions in Titan. Some further work reinforcing the link also in the results paragraph might be beneficial for the article.

*) Page 4. The asymmetric H-bond in which NH_3 species are involved (i.e., medium-to-strong accepting and weak donating) were reported not only in the Ref. 46 but also in other AIMD simulations (see, e.g., [JCP 136, 114509 (2012); JCP 161, 094503 (2024)]).

*) Page 4. When the Authors mention "We show that in solution, [...] hydrogen bonds from water." This was also anticipated in [JPCL 14, 7808 (2023)].

*) I was a bit confused by the EPSR simulation the Authors report starting from page 9. First of all, the acronym was never introduced. Also, some comments on the pros and cons, and the inherent limitations carried by this kind of simulation technique are mandatory to give to the reader some important reference to put in context the relative results. Are there any error bars in the H-bond enthalpies displayed in Fig. 6? If they are based on simple Lennard-Jones interactions, they should bring very large error bars since covalent bond deformation, large local polarization effects, and partial charge transfer are at the roots of the H-bond behavior in these systems, as outlined by a recent article comparing classical and ab initio simulations of water-ammonia solutions [JCP 161, 094503 (2024)].

*) Page 10. Cooperative H-bonding is a concept also partly anticipated in [JPCL 14, 7808 (2023)], where the role of local electric fields and polarized states is elucidated by first-principles.

Once all these points will be discussed and fixed, it will be my pleasure to reconsider the revised manuscript for eventual publication.

Version 1:

Reviewer comments:

Reviewer #1

(Remarks to the Author)

The authors considered most of my concerns replying in a pertinent way. I recommend publication of the work.

Reviewer #2

(Remarks to the Author)

The manuscript has been improved and my comments/questions have been satisfactorily addressed. I recommend the publication of the article after the small edit below:

L113-117: Add one sentence about the recent work by Leitner & Lunine (2019) who suggest that only up to about 5% of ammonia could be present in Titan's ocean

Leitner, M. A., & Lunine, J. I. (2019). Modeling early Titan's ocean composition. *Icarus*, 333, 61-70.

Reviewer #3

(Remarks to the Author)

I think that the paper, after the thorough revision made by the Authors, can now be accepted.

School of Physics and Astronomy
University of Leeds
Leeds, LS2 9JT
United Kingdom

p: +44 (0)113 34 38958
f: +44 (0)113 34 33900
e: l.dougan@leeds.ac.uk

UNIVERSITY OF LEEDS

2nd May 2025

Dear Editor,

Thank you for sending the reviews of our manuscript “**Solution Structure of Titan-Relevant Aqueous Ammonia by Neutron Diffraction**” which we submitted for consideration in Communications Chemistry. We have addressed all the points raised by the reviewers and updated the manuscript accordingly. Please find below our point by point response which includes all details of updates to the manuscript.

Thank you for considering this manuscript and we hope you agree that Communications Chemistry is the ideal home for this work.

Yours sincerely,

Lorna Dougan

A point-by-point response to the reviewer’s comments

Reference: COMMSCHEM-24-0729-T

Reviewer’s comments *italicised and numbered by reviewer.*

Our response: coloured in **bold**.

Reviewer 1:

Q1.1. *The application of neutron diffraction combined with ESPR analysis is robust, and the methodology is well-executed. The experimental setup is clearly described, and the data analysis provides a detailed characterization of the structural changes induced by ammonia.*

However while the neutron diffraction data analysis is thorough, the discussion is overly descriptive and lacks deeper insights into the underlying physical-chemical mechanisms driving the observed structural changes. The interpretation would benefit from exploring how ammonia influences hydrogen bonding networks and comparing these results with established models and previous literature. In this concerns the manuscript fails to adequately cite or build upon existing studies on the ammonia-water system under pressure. These studies provide critical insights into the behavior

of hydrogen bonding in mixed systems and could significantly enhance the interpretation of the results. Some selected examples are:

On the stability of the disordered molecular alloy phase of ammonia hemihydrate. CW Wilson, CL Bull, GW Stinton, DM Amos, ME Donnelly, JS Loveday. *The Journal of chemical physics* 142 (9)

Topologically frustrated ionisation in a water-ammonia ice mixture. C Liu, A Mafety, JA Queyroux, CW Wilson, H Zhang, K Béneut. *Nature communications* 8 (1), 1065

Melting curve and phase diagram of ammonia monohydrate at high pressure and temperature. H Zhang, F Datchi, LM Andriambarijaona, G Zhang, JA Queyroux. *The Journal of Chemical Physics* 153 (15)

Equation of state and electrical conductivity of warm dense ammonia at the conditions of large icy planets' interiors. JA Hernandez, A Ravasio, M Bethkenhagen, A Benuzzi-Mounaix. *AGU Fall Meeting Abstracts 2020, MR024-07*

Oriental Disorder Drives Site Disorder in Plastic Ammonia Hemihydrate. N Avallone, S Huppert, P Depondt, L Andriambarijaona, F Datchi, *Physical Review Letters* 133 (10), 106102

High pressure–temperature phase diagram of ammonia hemihydrate. L Andriambarijaona, F Datchi, H Zhang, K Béneut, B Baptiste. *Physical Review B* 108 (17), 174102

In point 1.1, the discussion is described as “overly descriptive and lacking deeper insight into the underlying physical-chemical mechanisms observed”, and reviewer 1 further comments that, “The interpretation would benefit from exploring how ammonia influences hydrogen bonding networks and comparing these results with established models and previous literature.”

To address how ammonia influences the hydrogen bond structure of water, we altered the Discussion to describe the role of the N_A-H_W hydrogen bond in organising the hydrogen bond structure of water by reference to the hydrogen bond enthalpies of: $N_{AMM}-H_{WAT}$, and O_W-H_W , in water molecules in the first solvation shell of ammonia, and in the surrounding bulk water respectively. This mechanism is now discussed in relation to the findings of Molecular Dynamics studies published by Munaò et al. in the *Journal of Chemical Physics* 161, 94503 (2024)⁶⁶ and Cassone et al. in *Journal of Physical Chemistry Letters* 14, 7808–7813 (2023)⁶⁸. We also discuss the role of the N_A-H_W hydrogen bond at higher ammonia concentrations.

The following replacement text is from the 3rd paragraph of the Discussion,

“It was anticipated that the ammonia molecule would exhibit an asymmetry in the strength of its hydrogen bond interaction water, and that this might curtail the extent of the hydrogen bonded water network. Our results confirmed this to be so, but unexpectedly we found that ammonia addition strengthens the enthalpy of the remnant water network, and both enhances the tetrahedral organisation of the water network, and the extent of large linear and ringed hydrogen bonded clusters. The result that ammonia addition leads to ice-likeness in its hydration shell is reminiscent of the predictions of Frank and Evans⁷¹ who had suggested that when a non-polar molecule dissolves in water, it passively modifies the surrounding molecular layers of water structure in the direction of greater crystallinity. In contrast, in aqueous 20.5 wt.% ammonia, at 273 K, our enthalpic analysis (Figure 6, Table S3) suggests that the strong N_A-H_W interaction ($-18.89 \text{ kJ mol}^{-1}$) influences the water network as its electronic effects are co-operatively translated through the ammonia hydration shell (H_W-O_W , $18.60 \text{ kJ mol}^{-1}$) into the bulk water network (H_W-O_W , $18.16 \text{ kJ mol}^{-1}$). That liquid water’s hydrogen bond environment can rapidly rearrange in response to solute addition is well known¹¹, and has been demonstrated by MD studies in aqueous ammonia⁵⁷. In this work we have highlighted the role of the N_A-H_W interaction in enhancing the tetrahedrality of water

molecules. The ice-like-ness of the water network in aqueous ammonia is made explicit in this work for the first time but MD studies have also highlighted the role of co-operative hydrogen bonding in modulating both networks. They also link the strength of the N_A-H_W bond to tetrahedral co-ordination of water molecules in ammonia's hydration shell⁵⁷, but in reverse, emphasising that it is the tetrahedrality of the hydrogen bonded water complex in ammonia's hydration shell that strengthens the N_A-H_W bond. They also suggest that as ammonia concentrations are increased the N_A-H_W hydrogen bond might weaken as electron pairs of the water molecules would be shared by an increasing number of H_A atoms⁵⁵."

The subject of our experiments is aqueous ammonia in conditions relevant to the NASA Dragonfly mission to the Selk impact crater at Titan's surface. We altered the text to remove any ambiguity, and the Introduction now opens with the comment that, "In 2034, the Dragonfly probe will land near the Selk impact crater on Saturn's moon Titan, a landing site selected for the likely presence of exposed deposits of water-rich material, potentially including materials where molten ice has interacted with organics¹, possibly in the presence of ammonia², whilst later in the work we comment that "The subject of this work is the intermolecular structure of aqueous ammonia in the context of the outer solar system and particularly the impact melts of Saturn's moon Titan"

We thank you for the recommended high-pressure articles and now cite Wilson et al, Liu et al, and Zhang et al. (the first three of the recommended high pressure works)...

"To understand the geophysics of the cores of the ice-giants, that models suggest are rich in ammonia, methane and water³, X-ray, and neutron diffraction studies have been performed on solid state ammonia hydrates under high pressures and a range of temperatures"³⁶⁻³⁸.

Point 1.1 concludes with a question about the ionisation of ammonia, "*In particular the possibility of ammonia ionization in presence of water should be taken into account.*"

We have added a section on partial ionisation in the Methods section...

"Partial Ionisation

Ammonia is a weak base that undergoes partial dissociation on addition to water: $NH_3 + H_2O \rightleftharpoons NH_4^+ + OH^-$. The base dissociation constant (k_b) is 1.77×10^{-5} at RTP⁷⁹, and 1.37×10^{-5} at STP⁸⁰. The degree of ionisation is concentration dependent, and is described by Ostwalds Dilution Law and Equation 6,

$k_b = \frac{\alpha^2}{1 - \alpha} c_0,$	(6)
--	------------

where α is the degree of dissociation, and c_0 is the molar concentration of ammonia added to solution. A 20.5 wt.% ammonia solution has a density of 0.92 g cm^{-3} ⁸², and a molarity of 11.07. Solving for α in Equation 6, the percentage of ionisation in 20.5 wt.% aqueous ammonia at RTP is 0.13%. Given this small value, ammonia ionisation has been ignored in the modelling of scattering from the aqueous ammonia samples."

The ionisation is similarly ignored by the classical MD and ab initio that we now cite (Munaò et al. in the *Journal of Chemical Physics* 161, 94503 (2024)⁵⁴). Another work by Weinhardt et al. on a slightly more concentrated ammonia solution, "Probing hydrogen bonding orbitals: resonant inelastic soft X-ray scattering of aqueous NH_3 , in PCCP, 2015 finds "that only a minor fraction (c.0.1%) of the NH_3 (ND_3) is protonated, forming NH_4^+ (ND_4^+)".

Q1.2. *The connection to the Dragonfly mission and Titan's Selk impact crater is timely and aligns with the growing interest in extraterrestrial aqueous chemistry. However, the planetary implications of the present study are rather speculative and somewhat overstretched. The chosen temperature and pressure conditions (1 bar and 273–298 K) are not directly relevant to Titan's subsurface ocean*

or atmospheric conditions, which are likely to involve either higher pressures or lower temperatures. The extrapolation to Titan's environment feels tenuous without a more direct connection to those conditions.

Our study pertains to the structure of aqueous ammonia under the conditions relevant to the interaction of organics and water at Titan's surface in the aftermath of an impact or cryovolcanic event. In the Abstract to the article 'Selection and Characteristics of the Dragonfly Landing Site near Selk Crater, Titan'³, the NASA mission science team outline that, "The factors contributing to the initial selection of a dune site near the Selk impact structure on Titan as the first landing site for the Dragonfly mission are described. These include arrival geometry and aerodynamic/thermodynamic considerations, illumination, and Earth visibility, as well as the likely presence of exposed deposits of water-rich material, potentially including materials where molten ice has interacted with organics." Similarly, planetary scientists such as McDonald and Sagan published work on 'Chemical Investigation of Titan and Triton Tholins' which considered the interaction of organics, liquid water and ammonia in the Titan regolith. The hydrolyses of organics (Titan-tholin simulants) that so interested McDonald and Sagan have been modelled in 5 experiments at conditions above the ammonia-water eutectic at 1 bar, and our experiment approximate these conditions (1 bar, 273 K, 298 K). To make these point clear to the reader, we open this work by referencing the NASA Dragonfly's selection of a landing-site at Titan's Selk impact crater, "In 2034, the Dragonfly probe will land near the Selk impact crater, a site selected because of the likely presence of exposed deposits of water-rich material, potentially including materials where molten ice has interacted with organics"³, possibly in the presence of ammonia⁴." We also comment later in the Introduction that, "The subject of this work is the intermolecular structure of aqueous ammonia in the context of the outer solar system and particularly the impact melts of Saturn's moon Titan."

We also cite details of Titan-tholin hydrolyses experiments with aqueous ammonia in the Introduction and we inserted Table S1 into the Supplementary Information that summarises the Temperature and Pressure conditions pertaining to the tholin-hydrolyses experiments. In the main text we comment that, "The experimental conditions are also similar to the experimental conditions of Titan-tholin hydrolyses in aqueous ammonia, Table S1^{4,47,62-64}". Table S1 is reproduced below.

Table S1. The intermolecular interactions of water molecules in aq. ammonia (273 K, 298 K), and pure water (298 K)

Conditions	Somogyi, 2005	Neish, 2009	Ramirez, 2010	Poch, 2012	Cleaves, 2014
Ammonia (wt.%)	1,2	13	3.125, 6.25,12.5,25	25	15
Main text citation	48	4	49	50	51
Temp (K), 1 bar	298,373	253, 273, 297	96, 253, 277	253, 279	253

1.3. *The rationale for selecting the specific 20.5 wt.% ammonia composition is not well articulated. While this may represent a eutectic-like mixture, its relevance to Titan's subsurface ocean or atmosphere is unclear. Stronger justification or supporting evidence is needed to establish this composition's pertinence to Titan's chemistry.*

We previously wrote that, "A 20.5 wt.% ammonia solution is broadly similar to conditions expected at a Titan impact melt, or cryolava, and concentrated enough to clearly observe the perturbing effects of the solute on the water structure with neutron diffraction", and we now supplement this by referencing the experimental conditions applying to 5 published hydrolyses of Titan-tholin simulant in aqueous ammonia that were referenced in Table S1 above."

1.4. Ammonia substantially perturbs the HBs structure of water, and this is manifest in the deep eutectic shown by the water-ammonia system. Such a kind of behavior has been observed also in salty aqueous solutions where the water structure was also investigated by neutron diffraction and ESPR simulations. A comparison between the ammonia addition and salt addition could be pertinent.

Thank you for the suggestion.

We re-wrote the Introduction and discuss the solvation shell in general, rather than in relation to a specific terrestrial aqueous system. We state in the Introduction that, “Terrestrial biochemistry has shown that the self-assembly and aggregation of molecular building blocks is linked to the hydrophobic interaction, an interaction that was historically linked to ‘quasi-solid’⁷ crystalline structure in the hydration layer of volatile solutes such as methane⁷, and in non-polar solute hydration shells in general⁸”

Previous iterations of the eutectic imaged in Figure 1, referenced the eutectics of aqueous NaCl and CaCl₂ but we considered this might be distracting for the reader. A comparison to a terrestrial salt solution, would be interesting, but we consider that in the space available, we need to prioritise a description of aqueous ammonia’s intermolecular structure and some explanation for the effects that ammonia-addition exerts on water structure.

Minor point:

-The indication of the sample temperature as a suffix to the chemical formula in figures is misleading.

Thank you. We amended Figure 2, to include a ‘K’ in the suffix.

-The term mimetic for neutron scattering studies should be explained.

Thank you. We altered the text slightly from “Mimetic neutron scattering studies of simplified solutions...”, to “Neutron diffraction studies of simplified mimetic solutions...” to indicate that the technique is neutron diffraction, and the sample is an idealised and simplified mimic of a Titan-relevant aqueous ammonia.

Reviewer 2:

Please see below a list of suggestions/questions that the authors might consider to improve the manuscript.

2.1 L89-93: There is increasing evidence suggesting that comets are the building blocks of outer solar system objects. Observations of comet 67P/Churyumov-Gerasimenko by the Rosetta mission imply that the abundance of ammonia is likely around ~1-2% relative to water [Mumma, M.J. and S.B. Charnley (2011). Annual Review of Astronomy and Astrophysics, 49: p. 471-524], rather than the ~5-15% predicted by early chemical models. I think this should be discussed here.

This is an important question. Very recently, Nature and Nature Astronomy featured articles on the sample analysis of the Bennu asteroid. The article in Nature Astronomy, ‘Abundant ammonia and nitrogen-rich soluble organic matter in samples from asteroid (101955) Bennu’, Glavin et al. (2025) is exciting because it suggests that complex organics formed in hydrous planetesimals and that they might later have been delivered to the inner solar system, and Earth, in impact events. We see this article as a complement to this literature. Our focus is the impact of ammonia on liquid water’s intermolecular structure which is relevant to the hydration of organics in ammonia-rich ocean worlds. We agree that it is important then to scope out the extent of aqueous ammonia in the outer solar system, and we have altered the Introduction to do this. We removed the discussion of Titan’s tholins, and focused on the extent of aqueous ammonia in the solar system. We opened our Introduction by referencing the discovery that the Bennu parent body appears to have evolved in ammonia-rich fluids,

and we go onto describe the distribution of ammonia in the outer solar system. We reproduce here the opening 2 paragraphs of our revised manuscript...

“In 2034, the Dragonfly probe will land near the Selk impact crater on Saturn’s moon Titan, a landing site selected for the likely presence of exposed deposits of water-rich material, potentially including materials where molten ice has interacted with organics³, possibly in the presence of ammonia⁴. The significance of the interaction of ammonia with liquid water, and organics, was recently highlighted by laboratory analysis of material returned from the surface of asteroid, Bennu⁵. Analysis of the Bennu sample found ‘abundant ammonia’, and evidence for soluble organics formation and alteration by low-temperature reactions, possibly in ammonia-rich fluids⁶. The organic matter included 14 proteinogenic amino acids, amines, formaldehyde, carboxylic acids, polycyclic aromatic hydrocarbons, and the five nucleobases found in DNA and RNA⁶. The presence of the ‘building blocks of Life’ with evidence of water-rich fluids has significance for the origins of Life on Earth. That these organic building blocks may have been solvated by aqueous ammonia underlines the importance of understanding the effect that ammonia addition has on the intermolecular structure of liquid water and pre-biotic molecules. Terrestrial biochemistry has shown that the self-assembly and aggregation of molecular building blocks is linked to the hydrophobic interaction, an interaction that was historically linked to ‘quasi-solid’⁷ crystalline structure in the hydration layer of volatile solutes such as methane⁷, and in non-polar solute hydration shells in general⁸. In the case of terrestrial structured biomolecules, such as proteins, protein folding is driven by the totality of atomic interactions in a solvation shell of hydrogen bonded water molecules, ions and solutes⁹. The properties of Earth’s liquid water that make it a ‘matrix for life’¹⁰ have been related to the strength and extent of what is the densest known hydrogen-bond network of any known material¹¹. The effect then of ammonia addition, a molecule capable of hydrogen bond formation, on the intermolecular structure of liquid water is then relevant to the assembly and stability of pre-biotic molecules in ammonia-rich ocean worlds, and the subsequent delivery of complex organics to the inner solar system. Figure 1b describes the phase diagram of ammonia-water mixtures at 1 bar that includes a deep eutectic at 33 wt.% ammonia, 176 K¹ implying that ammonia addition significantly perturbs the hydrogen bonded structure of liquid water.

The subject of this work is the intermolecular structure of aqueous ammonia in the context of the outer solar system and particularly the impact melts of Saturn’s moon Titan. In 1972, a theoretical model was proposed for the formation of the ice-giants and their icy satellites through the low temperature condensation of ices in thermodynamic equilibrium with the cooling gas of the solar nebula¹². The condensation of significant quantities of solid ammonia hydrates was a key prediction of this model such that the chemical composition of the icy satellites was estimated to be ~10% ammonia by mass¹³. The seminal work¹² led to astrobiological interest in the effect that ammonia might have in depressing the freezing point of water, a phenomenon that would extend the phase space of liquid water in ammonia-rich worlds. Much interest centred on Saturn’s moon Titan whose methane-rich atmosphere^{14,15} was thought to host photolytic chemistry and a red aerosol of complex hydrocarbons that would be hydrolysed to form amino acids in the presence of ammonia in Titan’s regolith¹⁶. In 2005, Cassini’s Huygen’s probe measured Titan’s atmosphere to be 98% nitrogen and 2% methane¹⁷ before landing on what appeared to be dry lakebed at 93.7 K, 1470 mbar¹⁸. Observations by the Cassini spacecraft¹⁹ characterised Titan as a volatile-rich world with methane lakes at its Northern pole²⁰, and a sub-surface ocean of liquid-water²¹ that might be enriched in ammonia²², although other solutes such as ammonium sulphate²¹ have also been proposed. Initial models of Titan’s sub-surface ocean were influenced by astronomic studies of the environment of Young Stellar Objects that measured ammonia ice abundances of up to ~15% with respect to water-ice²³. Despite the predictions of extensive reservoirs of ammonia hydrate condensate in the outer solar system^{12,24}, sample analysis and spectroscopic surveys suggest that ammonia seems to have survived only locally where it formed in stoichiometric excess^{1,25}. Chondritic meteorite and asteroids sample relatively

primitive unmelted condensate from the solar nebula²⁶ and they evidence significant variability in ammonia abundance. The carbonaceous meteorites of the Renazzo-type contain ammonia of up to 10% by mass of the insoluble organic material fraction, values far in excess of ammonia abundances in other carbonaceous meteorites²⁷ and the Ryugu asteroid⁶. Photometric and spectroscopic surveys of cometary nuclei suggest their ammonia concentrations are in the range of 0.2%-1.4% with respect to water abundance²⁸. The difference in the predictions of chemical models, and the results of sample analysis has been explained by the sequestration of ammonia by other volatiles and salts such as carbon dioxide, formaldehyde and magnesium sulphate to form ammonium salts¹. Visible-Infrared mapping of the surface of Ceres, a dwarf-planet in the main asteroid belt finds the signature of ammonia salts that are consistent with ammonia sequestration²⁹. Spectroscopic studies of Titan have proved difficult to interpret because its thick haze and cloudy atmosphere block the transmission of many visible and IR wavelengths, leaving only limited windows in which spectral observations can be made³⁰. Few compounds have been identified beyond contaminated water-ice³⁰. Infrared spectroscopy from the Mauna Loa observatory suggests the presence of ammonia hydrate, or flash frozen ammonia water, on the surface of the Uranian moons Miranda^{31,32} and Ariel³³. Cassini's Ion and Neutral Mass Spectrometer flew through Enceladus's ice-water plume in 2008 and measured a volume mixing ratio of 90 % water and 0.82 +/- 0.02 % ammonia³⁴ that the authors suggest is circumstantial support for the presence of ammonia in Titan."

2.2 L115-116: Why did you select a 20.5 wt% ammonia concentration? Does this concentration more accurately reflect the conditions expected in Titan's impact melt pools, or is it meant to represent an average of the concentrations you mentioned earlier (14wt% and 33wt%)? You mentioned that these conditions enable a study relevant to Titan, but could you clarify whether this specific concentration was chosen to better simulate the conditions of Titan's cryolava or for another reason?

We seek to uncover the intermolecular structure of an aqueous ammonia solution in conditions relevant to Titan's impact melt pools. We addressed this question by considering previous studies on the hydrolyses of a Titan-tholin simulant by aqueous ammonia solutions. These were all carried out at 1 bar, but with a range of temperatures and ammonia concentrations (see Table S1 appended to Question 1.2). Our experimental conditions are consistent with these experimental conditions and allow us to produce a reliable formative model of the intermolecular structure of ammonia. The following paragraph in the Introduction addresses the choice of ammonia concentration.

"A 20.5 wt.% ammonia solution is broadly similar to conditions expected at a cooling Titan impact melt, or cryolava, above the eutectic, and concentrated enough to clearly observe the perturbing effects of the solute on the water structure with neutron diffraction. The experimental conditions are similar to the experimental conditions of Titan-tholin hydrolyses in aqueous ammonia, Table S1 ^{4,47,62-64}".

2.3 How would ammonia behave as the temperature of the impact melt pool decreases at lower temperatures (just above the eutectic for example)?

We have not discussed this issue although it is apparent from our data that the tetrahedrality, and topology of the hydrogen bond network is strongly temperature sensitive. The effect of temperature is of course not an ammonia-effect.

2.4 How do your results scale to higher ammonia concentrations or higher pressures (800 MPa) as found at the bottom of Titan's ocean?

This is another interesting question. Jeffrey Kargel published work on ‘Rheological properties of ammonia-water liquids and crystal-liquid slurries: Planetological applications’ in *Icarus* (1991) which shows that aqueous ammonia behaves like honey as it is poured at conditions just above the eutectic whereas the 20.5 wt.% ammonia appeared like liquid water. To speak authoritatively on this we would need to repeat the experiment under eutectic-like conditions to make any predictions, however in the Discussion we now cite some discussion on the impact of the N_A-H_W bond which might be expected to weaken as ammonia-concentration increases. This Discussion is compatible with our results...

“The ice-like-ness of the water network in aqueous ammonia is made explicit in this work for the first time but MD studies have also highlighted the role of co-operative hydrogen bonding in modulating both networks. They also link the strength of the N_A-H_W bond to tetrahedral co-ordination of water molecules in ammonia’s hydration shell, but in reverse, emphasising that it is the tetrahedrality of the hydrogen bonded water complex in ammonia’s hydration shell⁵⁷ that strengthens the N_A-H_W bond. They also suggest that as ammonia concentrations are increased the N_A-H_W hydrogen bond might weaken as electron pairs of the water molecules would be shared by an increasing number of H_A atoms⁵⁵.”

2.5 Given that the results suggest ammonia significantly organizes the surrounding water network, could these findings provide insights into ammonia's interaction with methane clathrates on Titan, potentially favoring partial dissociation rather than the incorporation of ammonia into the clathrate structure?

This is another wonderful suggestion. We had not considered this question, but on consideration we see that this is a relevant question, and experimental work in the literature indeed suggests that ammonia destabilises methane clathrate structures. We have cited this work at the end of our Discussion...

“The organisation of water by ammonia, may have implications for the hydrophobic interaction, a significant force in molecular biology⁷². Historically, Kauzmann linked this interaction to enhanced structure in water around non-polar solutes like methane, or non-polar organic atoms⁸. Over the years, support for this theory waned⁷², as direct structural evidence for such quasi-crystalline states around hydrophobic atoms has been lacking⁷³. Some insight into the possible structure of Frank-Evans icebergs was provided by the structure of clathrate hydrates that are open crystalline networks of hydrogen bonded water networks that host small guest molecules such as methane. In aqueous ammonia, our experiments have shown that the nitrogen atom of ammonia organises a hydrogen bond network of water around it. This result leads to a hypothesis that ammonia might similarly perturb clathrate structure, and this result is compatible with high pressure experimental work on a 10 wt.% ammonia solution that has been shown to lower the dissociation temperature of methane clathrates by 14–25 K at pressures above 5 MPa⁷⁴. This is an interesting result because it implies that the ammonia molecule might generally disrupt the structure of liquid water around hydrophobes and thus reduce the extent of the hydrophobic effect around pre-biotic molecules with implications for the self-assembly of pre-biotic molecules in ammonia-rich ocean worlds. It remains to be seen just how prevalent aqueous ammonia is in the solar system’s icy moons, but the results of this experiment suggest that the ammonia molecule, in significantly organising the surrounding water network, has structural effects on water that may have consequences for the assembly of the building blocks of Life.”

Minor edits:

L165: It should be "the presence," not "the e presence."

L253: It should be "Figure 8c," not "S8c."

L413: There seems to be a typo with "figure S9."

All corrected.

Thank you for your comments!

Reviewer 3:

Dougan and co-workers report an interesting experimental investigation on the water structure in aqueous ammonia solutions. In particular, they show that ammonia addition leads to the formation of ice-like motifs in ammonia's hydration shell. Overall, the reported findings represent a valuable contribution to the community, even though some previous computational investigations already predicted an ammonia-induced structuring of the local water environment leading to important consequences in the reactivity of these acid-base systems [JPCL 14, 7808 (2023)]. In the following, I'm going to list a series of suggestions that the Authors could consider before submitting a revised version of the article for eventual publication.

3.1 *) The local water structuring induced by the addition of ammonia was predicted and reported by a series of *ab initio* molecular dynamics simulations of water-ammonia mixtures, mainly with the aim of disclosing the role of local molecular clustering in the formation of NH_4^+ and OH^- species [JPCL 14, 7808 (2023)]. In this contribution, the Authors show how the proton-accepting capability held by the N atom is capable of polarizing the H_2O H-bond donor molecule and, consequently, its hydration shell, leading to more structured local configurations. Although this represented only a computational prediction, due to the adherence of the results, it would be worth discussing these predictions also in the current investigation where, however, such a work is not mentioned.

Thank you for bringing this work, and related works (Munaò et al., J. Chem. Phys. 161, 094503 (2024), Bankura, A. & Chandra, A. Journal of Chemical Physics 136, 114509 (2012) to our attention. We now cite, in the Introduction, these 3 works in addition to the work by Ekimova et al. previously cited.,,

"Spectroscopic studies of water-ammonia mixtures in helium drops⁵³ and in aqueous solution⁵⁴, show evidence for an ammonia-water hydrogen bond asymmetry and this result has been replicated by classical and *ab-initio* molecular dynamics (MD) studies⁵⁴⁻⁵⁷."

In the Discussion we reference Munaò et al's, finding that tetrahedral complexes of water molecules, modulate, or strengthen, the $\text{N}_A\text{-H}_W$ bond...

"The ice-like-ness of the water network in aqueous ammonia is made explicit in this work for the first time but MD studies have also highlighted the role of co-operative hydrogen bonding in modulating both networks. They also link the strength of the $\text{N}_A\text{-H}_W$ bond to tetrahedral co-ordination of water molecules in ammonia's hydration shell⁵⁷, but in reverse, emphasizing that it is the tetrahedrality of the hydrogen bonded water complex in ammonia's hydration shell that strengthens the $\text{N}_A\text{-H}_W$ bond. They also suggest that as ammonia concentrations are increased the $\text{N}_A\text{-H}_W$ hydrogen bond might weaken as electron pairs of the water molecules would be shared by an increasing number of H_A atoms⁵⁵. The MD studies also highlighted the significance of the proton affinity of the nitrogen atoms of ammonia in aqueous ammonia solutions⁵⁵ and have quantitatively testified that the NH_3 species are more prone to accept protons than H_2O molecules⁵⁷. The MD results are compatible with our results and underline the role of co-operative hydrogen bonding in modulating hydrogen bond networks."

3.2 *) Title, abstract, introduction, and the conclusion paragraph link the reported findings with aqueous ammonia solutions present in Titan. However, the remainder part of the article (i.e., the

results paragraph) doesn't mention any plausible connections with what we know about ammonia aqueous solutions in Titan. Some further work reinforcing the link also in the results paragraph might be beneficial for the article. **Thank you for this feedback. This question is compatible with a suggestion from Reviewer 2 (2.5) who suggested we consider the effect of ammonia on the stability of methane clathrates. We have expanded the Discussion to cite experimental work that shows that ammonia addition destabilises methane clathrates. This work suggests that the effect of the N_A atom on hydrogen bonding in water molecules might be relevant to clathrates as well as liquid water structure.**

3.3 *) Page 4. The asymmetric H-bond in which NH₃ species are involved (i.e., medium-to-strong accepting and weak donating) were reported not only in the Ref. 46 but also in other AIMD simulations (see, e.g., [JCP 136, 114509 (2012); JCP 161, 094503 (2024)]).

*) Page 4. When the Authors mention "We show that in solution, [...] hydrogen bonds from water." This was also anticipated in [JPCL 14, 7808 (2023)].

Thank you for highlighting this literature. As above (3.2, 3.3) we now cite these findings,

"Spectroscopic studies of water-ammonia mixtures in helium drops⁵³ and in aqueous solution⁵⁴, show evidence for an ammonia-water hydrogen bond asymmetry and this result has been replicated by classical and *ab-initio* molecular dynamics (MD) studies⁵⁴⁻⁵⁷."

3.4 *) I was a bit confused by the EPSR simulation the Authors report starting from page 9. First of all, the acronym was never introduced. Also, some comments on the pros and cons, and the inherent limitations carried by this kind of simulation technique are mandatory to give to the reader some important reference to put in context the relative results. Are there any error bars in the H-bond enthalpies displayed in Fig. 6? If they are based on simple Lennard-Jones interactions, they should bring very large error bars since covalent bond deformation, large local polarization effects, and partial charge transfer are at the roots of the H-bond behavior in these systems, as outlined by a recent article comparing classical and *ab initio* simulations of water-ammonia solutions [JCP 161, 094503 (2024)].

The acronym EPSR s introduced in the Introduction ("This study uses neutron diffraction with isotopic substitution (NDIS), allied to computer modelling of the diffraction datasets using Empirical Potential Structure Refinement (EPSR)...)" and to help address such questions the EPSR procedure is further detailed in the Methods section.

We slightly altered the Introduction to make it clear that this is a structural refinement system, used to interpret measured diffraction data. We also altered the text in the Methods section to indicate that the structural atomic solution is one that is consistent with the measured structure factor derived from the measured diffraction datasets and not necessarily a unique solution that matches the sample structure...

"In this way, the pair potentials are perturbed by EP terms, derived from the diffraction measurements, and the structure in the simulation box is refined so that it is consistent with the measured structure factor derived from the measured diffraction datasets. It is important to note that the solution is compatible with the measured structure factor but that it may not reproduce the exact atomic structure of the measured solution."

In respect to the hydrogen bond interaction energies detailed in Figure 6, these are Gaussian fits to hydrogen bond interactions derived from random sampling of the atomic model configurations of the samples that have been refined by the diffraction data, and the variance is apparent in the Gaussian fits. Table S3 tabulates the mean hydrogen bond enthalpies evaluated from the spatial configuration of hydrogen bonded complexes, and their interatomic

Lennard-Jones and coulomb potentials using the reference potentials for each atom-type, and the standard error.

3.5 *) Page 10. Cooperative H-bonding is a concept also partly anticipated in [JPCL 14, 7808 (2023)], where the role of local electric fields and polarized states is elucidated by first-principles.

Thank you for highlighting this work. We amended our Discussion of our results, and this text from the 4th paragraph in the discussion is relevant:

“That liquid water’s hydrogen bond environment can rapidly rearrange in response to solute addition is well known¹¹ and has been demonstrated by MD studies in aqueous ammonia⁵⁷. In this work we have highlighted the role of the N_A-H_W interaction in enhancing the tetrahedrality of water molecules. The ice-like-ness of the water network in aqueous ammonia is made explicit in this work for the first time but MD studies have also highlighted the role of co-operative hydrogen bonding in modulating both networks. They also link the strength of the N_A-H_W bond to tetrahedral co-ordination of water molecules in ammonia’s hydration shell⁵⁷, but in reverse, emphasising that it is the tetrahedrality of the hydrogen bonded water complex in ammonia’s hydration shell that strengthens the N_A-H_W bond. They also suggest that as ammonia concentrations are increased the N_A-H_W hydrogen bond might weaken as electron pairs of the water molecules would be shared by an increasing number of H_A atoms⁵⁵. ”

The manuscript explores the structural changes induced by the addition of ammonia to water, focusing on a 20.5 wt.% ammonia-water solution at two temperatures, 273 K and 298 K, and 1 bar pressure. By employing neutron diffraction and utilizing empirical structure refinement (ESPR) analysis, the study observes enhanced tetrahedral ordering and ice-like motifs in the ammonia hydration shell. These findings are linked to potential implications for Titan's aqueous chemistry and ammonia-rich ocean worlds, providing context for NASA's upcoming Dragonfly mission.

I have the following comments:

The application of neutron diffraction combined with ESPR analysis is robust, and the methodology is well-executed. The experimental setup is clearly described, and the data analysis provides a detailed characterization of the structural changes induced by ammonia. However while the neutron diffraction data analysis is thorough, the discussion is overly descriptive and lacks deeper insights into the underlying physical-chemical mechanisms driving the observed structural changes. The interpretation would benefit from exploring how ammonia influences hydrogen bonding networks and comparing these results with established models and previous literature. In this concerns the manuscript fails to adequately cite or build upon existing studies on the ammonia-water system under pressure. These studies provide critical insights into the behavior of hydrogen bonding in mixed systems and could significantly enhance the interpretation of the results. Some selected examples are:

On the stability of the disordered molecular alloy phase of ammonia hemihydrate

CW Wilson, CL Bull, GW Stinton, DM Amos, ME Donnelly, JS Loveday

The Journal of chemical physics 142 (9)

Topologically frustrated ionisation in a water-ammonia ice mixture

C Liu, A Mafety, JA Queyroux, CW Wilson, H Zhang, K Béneut, ...

Nature communications 8 (1), 1065

Melting curve and phase diagram of ammonia monohydrate at high pressure and temperature

H Zhang, F Datchi, LM Andriambarijaona, G Zhang, JA Queyroux, ...

The Journal of Chemical Physics 153 (15)

Equation of state and electrical conductivity of warm dense ammonia at the conditions of large icy planets' interiors.

JA Hernandez, A Ravasio, M Bethkenhagen, A Benuzzi-Mounaix, ...

AGU Fall Meeting Abstracts 2020, MR024-07

Orientational Disorder Drives Site Disorder in Plastic Ammonia Hemihydrate

N Avallone, S Huppert, P Depondt, L Andriambarijaona, F Datchi, ...

Physical Review Letters 133 (10), 106102

High pressure–temperature phase diagram of ammonia hemihydrate

L Andriambarijaona, F Datchi, H Zhang, K Béneut, B Baptiste, ...

Physical Review B 108 (17), 174102

High pressure–temperature phase diagram of ammonia hemihydrate

L Andriambarijaona, F Datchi, H Zhang, K Béneut, B Baptiste, ...

Physical Review B 108 (17), 174102

In particular the possibility of ammonia ionization in presence of water should be taken into account.

The identification of enhanced tetrahedral ordering in ammonia's hydration shell provides an intriguing perspective on the perturbation of water's intermolecular structure by ammonia.

2. The connection to the Dragonfly mission and Titan's Selk impact crater is timely and aligns with the growing interest in extraterrestrial aqueous chemistry. However, the planetary implications of the present study are rather speculative and somewhat overstretched. The chosen temperature and pressure conditions (1 bar and 273–298 K) are not directly relevant to Titan's subsurface ocean or atmospheric conditions, which are likely to involve either higher pressures or lower temperatures. The extrapolation to Titan's environment feels tenuous without a more direct connection to those conditions.

3. The rationale for selecting the specific 20.5 wt.% ammonia composition is not well-articulated. While this may represent a eutectic-like mixture, its relevance to Titan's subsurface ocean or atmosphere is unclear. Stronger justification or supporting evidence is needed to establish this composition's pertinence to Titan's chemistry.

4. Ammonia substantially perturbs the HBs structure of water, and this is manifest in the deep eutectic shown by the water-ammonia system. Such a kind of behavior has been observed also in salty aqueous solutions where the water structure was also investigated by neutron diffraction and ESPR simulations. A comparison between the ammonia addition and salt addition could be pertinent.

Minor point:

- The indication of the sample temperature as a suffix to the chemical formula in figures is misleading
- The term mimetic for neutron scattering studies should be explained

While the study presents robust experimental work and intriguing findings, its current form does not meet the interdisciplinary depth and chemical-p relevance expected for *Communications Chemistry*. I recommend major revisions to address the discussion's depth, incorporate relevant literature, and refine the extrapolation to Titan's environment. Alternatively, the authors may consider targeting a specialized journal in physical chemistry.

Given these substantial concerns, I recommend that the manuscript undergo major revisions and be submitted to a more specialized journal.